# Deep Learning to Unveil Correlations between Urban Landscape and Population Health [note 1]

**DOI:** 10.3390/s20072105

**Published:** 2020-04-08

**Authors:** Daniele Pala, Alessandro Aldo Caldarone, Marica Franzini, Alberto Malovini, Cristiana Larizza, Vittorio Casella, Riccardo Bellazzi

**Affiliations:** 1Department of Electrical, Computer and Biomedical Engineering, via Ferrata 5, 27100 Pavia, Italy; alessandroaldo.caldarone01@universitadipavia.it (A.A.C.); cristiana.larizza@unipv.it (C.L.); riccardo.bellazzi@unipv.it (R.B.); 2IRCCS ICS Maugeri, via S. Maugeri 2, 27100 Pavia, Italy; alberto.malovini@icsmaugeri.it; 3Department of Civil Engineering and Architecture, via Ferrata 3, 27100 Pavia, Italy; marica.franzini@unipv.it (M.F.); vittorio.casella@unipv.it (V.C.)

**Keywords:** transfer learning, deep learning, urban landscape, health indexes, public health, convolutional neural networks

## Abstract

The global healthcare landscape is continuously changing throughout the world as technology advances, leading to a gradual change in lifestyle. Several diseases such as asthma and cardiovascular conditions are becoming more diffuse, due to a rise in pollution exposure and a more sedentary lifestyle. Healthcare providers deal with increasing new challenges, and thanks to fast-developing big data technologies, they can be faced with systems that provide direct support to citizens. In this context, within the EU-funded Participatory Urban Living for Sustainable Environments (PULSE) project, we are implementing a data analytic platform designed to provide public health decision makers with advanced approaches, to jointly analyze maps and geospatial information with healthcare and air pollution data. In this paper we describe a component of such platforms, which couples deep learning analysis of urban geospatial images with healthcare indexes collected by the 500 Cities project. By applying a pre-learned deep Neural Network architecture, satellite images of New York City are analyzed and latent feature variables are extracted. These features are used to derive clusters, which are correlated with healthcare indicators by means of a multivariate classification model. Thanks to this pipeline, it is possible to show that, in New York City, health care indexes are significantly correlated to the urban landscape. This pipeline can serve as a basis to ease urban planning, since the same interventions can be organized on similar areas, even if geographically distant.

## 1. Introduction

With the recent technological advances and new societal landscape of the 21st century, new health challenges are emerging throughout the world. Most of the population today resides in big cities and this trend is expected to continue in the next decades [1], leading healthcare providers to face new problems, due to increasing air pollution and a changed lifestyle. Pathologies such as respiratory and cardiovascular diseases are increasing in all the western world, enlightening the necessity of targeted interventions [2,3]. These diseases are all the result of a heterogeneous combination of environmental and social exposures, sometimes difficult to identify, especially in an urban environment in which conditions can change significantly from one neighborhood to another [4].

In this contest, the PULSE (Participatory Urban Living for Sustainable Environments) project, funded by the European Commission, aims at developing a set of models and technologies, to predict and manage public health problems in cities and to promote health. It is based on a participatory approach, where citizens provide data through personal devices that are integrated with information from heterogeneous sources: open data, healthcare indexes, urban sensors and satellites. The project deals with various issues concerning air quality, lifestyle and personal behavior and it aims to investigate the correlations between exposure to atmospheric pollutants, the citizen habits and the health of the citizen themselves, focusing on asthma and type 2 diabetes. PULSE is being implemented in 7 major cities all over the world: Barcelona, Birmingham, New York, Paris, Pavia, Singapore and Keelung. Within PULSE, we are implementing a data analytic platform that will provide public health decision makers with advanced approaches to jointly analyze maps and geospatial information with healthcare data and air pollution measurements.

PULSE is creating a multi-technological platform made of several parts, from a personal app for the citizens to a set of geographical tools (WebGIS, sensor networks, spatially enabled analytics, etc.) to a dashboard for the Public Health Operators (PHOs), connecting citizens and policy makers in a unique comprehensive system.

Besides, recent advances in machine learning and deep learning enable the design and implementation of novel data analysis pipelines that allow fusing heterogeneous data sources to extract novel insights and predictive patterns [5,6]. These approaches seem particularly suitable to increase our insights in the relationships between the urban landscape of cities and the behavior of their residents, with particular focus on well-being and healthcare indexes. These tools may provide an effective instrument to ease the planning of intervention strategies, to aid localized health and social criticalities in the urban environment, reducing their time and costs. In this context, it can be of interest to health care planners and city decision-makers to have instruments able to find clusters of city areas that share similar urban structures and to analyze some behavioral indexes of their residents, in particular to see potential correlations and to plan similar interventions in the different clusters, even if such clusters refer to areas that are geographically far away.

In this paper we will describe the results obtained with a component of this platform, designed to couple deep learning analysis of geospatial images of cities and some healthcare and behavioral indexes. Our prototype was developed on New York City data, and shows that in this city, arguably one of the most heterogeneous urban sites in the world, the urban landscape significantly correlates with health indexes such as access to healthcare services, healthy behaviors, prevention of diseases and prevalence of various pathological conditions.

Our study is based on the transfer learning paradigm, i.e., the application of knowledge gained during the resolution of a specific problem to another problem, related to the first one [7]. In particular, we designed an analysis pipeline that applies a deep convolutional neural network algorithm already learnt on generic images, to a set of preprocessed satellite images of the city, categorizing them with respect to their urban landscape, i.e., presence of green areas, large building, roads, and residential areas. Then, through a suitable machine learning analysis, we were able to extract a series of correlations between the results of this algorithm and some health indexes associated with each specific geographical area.

This paper represents an extension of a previous work presented at the 17th International Conference on Smart Living and Public Health (iCOST), held in New York City in October 2019. The previous version is currently published in the conference proceedings [8]. These proceedings discuss how artificial intelligence, technology and the internet of things can impact public health, collecting a series of works presenting innovative projects. In 2019, PULSE was one of the main sponsors of this conference, and several of its features are described in the proceedings.

### Background and Related Work

The link between environment and health is well documented, and some literature focused on urban areas, health and environment has been recently published. For instance, Krefis et al. wrote a systematic review in 2018 [9], showing that the link between green areas and health have been addressed by several studies, but also pointing out a lack of interdisciplinary studies that approach the complexity of urban structure, its dynamics and links with wellbeing. Some socio-technical studies have been performed as well, such as the one conducted by Tavano Blessi et al. in Milan, Italy [10], in which they analyzed survey data to determine the influence of urban green areas in the precepted wellbeing of the population. In addition, deep learning has been already used in a number of studies on these topics, like for example the one by Helbich et al. [11], where they used deep learning on street view images to determine whether there was an association between the presence of green/blue areas and geriatric depression. Their results support this hypothesis, even though causal relationships were not fully investigated.

These studies are either very general (i.e., they focus on general wellbeing) or very specific, since they investigate the links between a specific environmental factor and a specific condition. In this paper, we present an analysis pipeline to try to answer the same research questions, but investigating the influence of urban structures in a series of preventable health complications, prevention strategies and behavioral health risk factors that mirror the association between the social structure of the city and the physical urban environment. To this end, we resorted to the capability of deep neural models to process images and to correlate them with outcome measurements, such as health indexes.

Deep neural models provide flexible instruments to perform the non-linear approximation of a variety of multivariate functions and to extract latent variables from a data set. In a nutshell, deep neural models are neural networks with many layers, able to map non-linear functions with a number of parameters that is typically lower than their equivalent models with one layer only, while preserving the same approximation accuracy. Such models are particularly attractive since they can be used to perform clustering, regression and classification starting from data sets made of images, texts and signals.

In dependence of the nature of the input data set, different architectures can be exploited, ranging from the combination of many convolutional layers in the case of images, to the use of long-term/short-term networks in the case of time series and speech/text data.

Recently, an increasing number of papers are using deep learning to examine the relationships between the urban landscape and some environmental data, as well as with citizens’ behavioral data [11,12,13,14].

One of the main limits of deep learning models is related to the large data sets needed to reliably estimate their parameters. In fact, in order to be able to gain the advantage of their capability of encoding even the finest details that can be important to map input data, large data sets are necessary in order to avoid getting trapped into noise resulting in overfitting and poor parameters estimates.

Rather interestingly, in order to deal with this problem, it is possible to resort to an increasing set of pre-trained deep learning models that can be used for the task of transfer learning [15], i.e., models that are able to represent the input space into a set of latent variables on the basis of a mapping mechanism, usually a deep neural network, learned on a large (external and independent) data set, so that the relationships between such latent variables and the outcomes can be later learned on a specific and smaller data set. A well-known example is Inception-v3, a convolutional neural network trained on more than a million images from the ImageNet database (http://www.image-net.org). The network has 48 layers and can classify images into one thousand object categories, including trees and many animals. Another interesting example is represented by the Painters [16] networks, developed to automatically classify paintings of famous artists. In principle, any of those methods can be used following the transfer learning paradigm, to represent images coming from the urban landscape of New York City (NYC).

## 2. Materials and Methods

### 2.1. Data

Our analysis is based on two data sources: NYC high resolution images and healthcare data coming from the 500 cities project [17]. NYC images have been collected by The National Agriculture Imagery Program (NAIP), that acquires aerial imagery during the agricultural growing seasons in the continental United States. In particular, we have downloaded an image of NYC having an original resolution of 0.5 m and have downsampled it to 2 m, which allows one to have a fine-grained representation of the aerial urban landscape (see Figure 1).

As will be explained in the following, the reason for the downsampling is that the original large image has been subdivided into tiles and the neural network adopted can accept images having a maximum size of 299 pixel; we had to tune the ground resolution in order to have meaningful tiles, embracing a sufficiently-sized area.

Health care data have been extracted from the repository made available by the 500 Cities project. “500 cities” is a collaboration between CDC, the Robert Wood Johnson Foundation, and the CDC Foundation (https://www.cdc.gov/500cities/index.htm). The project provides estimates for chronic disease risk factors (unhealthy behaviors), health outcomes, and clinical preventive service use for the largest 500 cities in the United States. Such estimates are provided for each census tract of a city.

Each American state is divided into counties, and the area of each county is further organized into census tracts. Census tracts are conventional geographical entities within the US counties [18] defined for census operations, and they represent the smallest territorial entity for which population data is available [19].

The last census in the United States was organized in 2010 and the number of census tracts which the American territory is divided into is 74,134. Generally, each of them is different from the other with respect to several features such as the population (between 1200 and 8000 people) and the spatial dimension, which depends on the area density [20].

Figure 2 shows an image of a census tract of Manhattan, while Figure 3 represents the city of New York divided into its census tracts, that, to be precise, are 2166.

The 24 chronic diseases measures provided by the project are listed in Table 1. They include major risk behaviors that lead to illness, suffering, and early death related to chronic diseases and conditions, as well as the conditions and diseases that are the most common, costly, and preventable of all health problems.

CDC provided estimates of the prevalence of each of the indicators listed in Table 1, both at the city level and at the census tract level.

For instance, the measure “Arthritis among adults aged >= 18 years” indicates the prevalence of people aged 18 or older who have been diagnosed with arthritis by a doctor, a nurse or another health specialist.

Most of these estimates were obtained by the CDC through telephone surveys to the US citizens. In this project, the estimates calculated on the whole city of NY were not used. Instead, we used only those obtained on each of its census tract instead, in order to investigate the hidden patterns between the structure of the city and health and behavior of its residents. It should be noted that in our analyses we tested 25 variables, since the indicator “Older adults aged ≥ 65 years who are up to date on a core set of clinical preventive services by age and sex” in our data was split into two separate variables based on sex. These variables are indicated with the acronyms *COREM* and *COREW*, respectively.

These estimates belong to the category of small areas estimates: this definition refers to statistical methods used to produce extremely precise estimates for very small geographical areas, where the size of the samples is minimal or even zero [21].

The importance of these “small area” epidemiological data lies in the fact that they can be used to develop and implement effective and targeted prevention services, identify rising health problems or control objectives of fundamental importance for health.

### 2.2. The Data Analysis Pipeline

The pipeline implemented in our work is described in Figure 4. The NAIP NYC image has been subdivided into image square blocks having sizes of 256 × 256 pixels, corresponding to a 512 m edge. It was therefore possible to estimate the value of each of the 24 variables collected by “500 Cities” in each block. During this process, blocks out of the tracts or over the sea have been excluded, thus reducing the dataset. The images have been then processed by a pre-trained deep model, thus extracting the final features for each image. Images are clustered by resorting to k-means clustering, and the clusters (also interpreted with visual inspection) have been associated to the healthcare indexes by statistical analysis.

### 2.3. Image Blocks

The NAIP NYC image has been subdivided into sub-images of 256 × 256 pixels. Each image is a square with an edge equal to 512 m.

First, it was determined by how many rows and columns of size 256 × 256 the NAIP raster image of NY is composed. The raster image can be represented as a matrix of size 23,972 × 23,892 × 3, where three indicates the number of channels (RGB) of the image and 23,972 are the number of rows and columns, respectively. Therefore, this matrix can be subdivided into a grid of 94 rows and 94 columns of size 256 × 256 pixels. After this subdivision, we obtained 8836 images with dimensions 256 × 256 × 3, and we saved all of them in a sequential order starting from the top left corner of the NAIP image.

It must be noted that the original image is georeferenced, meaning that each single pixel is precisely located in space. Indeed, digital images must be thought as mosaics composed by regular elements called pixels; they can be rectangular or, more commonly, squared. A georeferenced raster image is usually associated with a so-called world file [22].

In the most common implementation of image georeferentiation, images are generated so that pixel row and columns are parallel to the x and y axis of the geographical coordinates adopted. The world file basically supplies the coordinates of the center of the top-left pixel and pixel size, in geographical coordinates. By using the simple information mentioned, it is possible to precisely locate in space each single pixel constituting the image. Moreover, when sub-images are extracted, they can also be georeferenced by applying the same principles and a new world file can be generated for each of them.

The small derived tiles are georeferenced as well and can be effectively overlapped to the health and well-being maps. The images have been processed resorting to the MATLAB Image Processing and Mapping toolboxes, which are capable of properly managing georeferenced images. Figure 5 shows some examples of the resulting images.

It can be noticed that some of the tiles have white areas that correspond to unmapped zones, i.e., zones outside of the city boundary, including the ocean and the rivers. Due to the availability of the vector map of the borders of NYC, we have been able to quantify, for each tile, the amount of its surface lying inside the borders of the city; we then filtered the original tile set and maintained only those having a minimal overlapping of 90%.

### 2.4. Estimation of Healthcare Indexes

The healthcare indexes of the 500 Cities database are collected for census tracts and NYC has 2166 tracts. In order to carry out our analysis, we had to determine the value of the considered variables for each image block. In fact, a given tile overlaps, in general, several tracts. Therefore, we had to implement a simple estimator of the healthcare index of the block, as:
$$hci\left(block\right)=\frac{\sum_{j}w_{j}hci\left(j\right)}{\sum_{j}w_{j}}$$
where *hci*(*j*) is the value of the generic health care index for the j-*th* census tract and *w_j_* is the percentage of the image block covered by the mentioned tract. An example is shown in Figure 6 and Figure 7.

In order to properly quantify *hci*s, the blocks with a white area greater than 10% of the image have been removed. The final number of image blocks used for the following analysis has thus lowered to 2512 images. Each image has been then processed by resorting to a deep neural model to extract a set of latent features.

### 2.5. Deep Neural Network Processing and Clustering

Land use and land cover (LULC) analyses are usually performed with a large variety of methods, usually based on machine learning algorithms such as support vector machines and random forest, often adapted for image analysis. Image classification or clustering can be performed mainly with two different approaches: a pixel-based approach and a tile-based approach. In the first case, the whole image is subdivided into areas that are categorized according to the features of the pixels (e.g., RGB levels), whereas in the tile-based analysis, the image is divided into several sub-images that are classified separately. In our study, our main interest was to cluster the city neighborhoods according to their urban structure, so we decided to apply a tile-based approach and we selected a deep learning based analysis, since past research shows that in this approach, deep learning algorithms outperform the majority of traditional LULC methods [23,24].

As a deep neural network model used for transfer learning, we have selected the network developed for the 2016 Painters by number competition [16]. In such a competition, the goal was to learn how to discriminate the authors of paintings between 1584 unique painters and starting from a training set of 79,433 images; the test set was composed of 23,817 images. In this case, a deep neural network model was learned, with 23 layers, mostly convolutional layers with some max pooling layer. The Painters network computes a layer of 2048 latent variables before the final discrimination layer implemented with a soft-max non-linear function. Those latent variables can be used as a way to embed generic images in the latent space. Therefore, using the software Orange (**https://orange.biolab.si**) and its Python pipeline, we have processed all image blocks with the Painters model, thus obtaining a final data matrix of 2512 examples, with 2048 features.

This neural network was selected after testing all those made available by Orange (6 different CNNs with different structures, i.e., Inception v3 [25], VGG-16 [26], VGG-19 [27], Painters, DeepLoc [28], Openface [29]), using the t-SNE algorithm. This algorithm performs a dimensionality reduction, projecting multidimensional data into a 2D space, grouping the observation based on their possible similarities in the original space [30]. We tested each neural network and measured their capability of grouping together images with a high percentage of green color, and the results show that the Painters network was the best performing one. In particular, the features generated by the different deep learning models were mapped onto a bidimensional map using the t-SNE algorithm. Then, the samples with the largest percentage of green color were selected and the centroid of these samples in the t-SNE space were computed. Finally, the sum of squared Euclidean distances (SSE) of those samples with the centroid were computed, and the deep network architecture with the lowest SSE were thus selected. Let us note that the information about the percentage of green color of each tile was used only to compare deep learning architectures. It was not used in the following steps of the analysis.

Such features have been used to cluster the image blocks by resorting to the well-known K-means clustering algorithm. The value of K has been derived with a grid search between 2 and 6 and taking the value that maximizes the Silhouette coefficient.

### 2.6. Correlation and Statistical Analysis

The final step of the data analysis pipeline is represented by the search of statistical correlations between the clusters and their *hci*s. Univariate multinomial logistic regression was applied to estimate the probability to belong to a specific cluster given single variables’ values, i.e., to forecast if an image belongs to a cluster on the basis of the healthcare index only. Multivariate multinomial logistic regression was performed after the removal of samples characterized by missing values. A backward stepwise selection procedure based on AIC was applied, to identify the most informative set of variables jointly modulating the probability to belong to the clusters. Multinomial logistic regression and the stepwise selection procedure were implemented in the R packages “nnet” and “stats”, respectively. Analyses were performed by the R software tool version 3.5.1 (http://www.r-project.org).

## 3. Results

### 3.1. Clustering

K-means was run on the 2512 instances with Euclidean distance and 10 reruns. Four clusters were found to maximize Silhouette coefficient. The output of the clustering algorithm has been validated by analyzing the cluster distribution with the tSNE (t-distributed stochastic neighbor embedding) two-dimensional mapping, as reported in Figure 8. It is easy to see that the four clusters are in general well separated in the two-dimensional space. It is worthwhile mentioning that this criterion was qualitatively used to also assess other deep neural networks models; painters turned out to generate the clusters that had the best tSNE spatial distribution of clusters.

Thanks to visual inspection, it is possible to highlight that the four clusters correspond well to different urban landscapes. Cluster C1 corresponds to green areas, Cluster C2 to residential areas with small houses, Cluster C3 to industrial areas and larger buildings, Cluster C4 to residential with larger buildings. Some examples are shown in Figure 9 and Figure 10. Cluster analysis clearly shows that the deep neural network model is able to map images in the latent space that share the intuitive notion of similarity that humans may use when they have to classify urban landscape. The method is thus able to automatically cluster similar areas where similar interventions can be planned.

Since embedding through CNNs could be sensitive to adversarial image rotation [31], we selected four images for each cluster and repeated the procedure on rotated copies of these images, in order to verify whether our algorithm was invariant to rotation. The images were randomly rotated with angles of 90°, 180° and 270° and then processed by the same algorithm. The clustering results were then compared with the ones of the original images. Figure 11 shows the dendrogram resulting from the hierarchical clustering of the original images and the rotated ones. From these results, it could be noticed that the original images and rotated ones are clustered together in pairs, thus our approach is insensitive to this type of image rotation, which is the one relevant for our application.

Once the four clusters were found, we performed a few statistical analyses to investigate whether they could be correlated with the health indexes of the 500 Cities database. First of all, we performed a Chi-squared test over all the 25 variables in order to verify that they could be related to the clusters in a statistically significant way. In order to perform this test, which works on categorical variables, all the continuous variables of the 25 indicators were discretized with the following procedure: each variable was subdivided into three categories so that each category contains 1/3 of the total number of observations; after that, a contingency table with dimension 4 × 3 was created (where four is number of clusters and three are the categories), indicating for each cluster the number of observations that are within the ranges defined by the thresholds obtained with the discretization of the variables. Figure 12 shows a representation of this procedure.

For all 25 variables, the null hypothesis of independence between the clusters and the distribution of the health indexes could be rejected with a p-value lower than 0.05, thus confirming a statistical association between the clusters and the health indicators. Figure 13 shows a bar diagram representing the negative logarithm of the p-value (y axes) for each of the 25 variables (x axis).

The contingency tables obtained before the Chi-squared tests can also be visually represented as in Figure 14, where the distribution of the observations in each category for each cluster is represented with colors for the variable CHOLSCREEN, i.e., the percentage of adults older than 18 that have been screened for high cholesterol levels. In this image, it appears that the propensity of adults to get screened is proportional to the quantity of green areas of their neighborhood of residence.

Further tests were performed using a set of multinomial logistic regression models treating the four clusters as classes. Two different strategies were used: first, a univariate logistic regression model was fitted for each variable, then a multivariate approach was used to identify the most informative subset of variables with respect to the class taking into account their cross-dependencies.

Of course, being the logistic regression a binomial model, a model built on four classes is actually the result of three sub-models that compare the classes in pairs, using one level as baseline. In the context of the analyses presented, Cluster 1 (C1) was considered the baseline class value, while Cluster 2 (C2), Cluster 3 (C3) and Cluster 4 (C4) were the references.

Results from univariate multinomial logistic regression on continuous variables are reported in Table 2 and show that all variables tested were significantly associated with at least one cluster. The 10 variables showing the strongest statistical association with at least 1 cluster were COREW_Crud, CHOLSCREEN, COREM_Crud, CANCER_Cru, DENTAL_Cru, ACCESS2_Cr, MHLTH_Crud, PAPTEST_Cr, LPA_CrudeP and COLON_SCRE. Of these, COREW_Crud, CHOLSCREEN, COREM_Crud, CANCER_Cru, DENTAL_Cru, PAPTEST_Cr and COLON_SCRE were characterized by significantly lower values in C2, C3 and C4, compared to C1. Individuals with high values of these variables were less likely to belong to C2, C3 and C4, considering C1 as baseline (OR < 1, p-value < 0.01). On the opposite, subjects with high values of ACCESS2_Cr, MHLTH_Crud and LPA_CrudeP were more likely to belong to C2, C3 and C4 compared to C1 (OR > 1, p-value < 0.001).

Analogously, all discretized variables were differentially distributed in at least one reference cluster compared to cluster 1 (p < 0.05), as shown in Table 3. Most of the 10 variables showing the strongest statistical association with clusters overlapped with the top 10 variables associated with continuous distributions, with eight attributes (CHOLSCREEN, COREW_Crud, COREM_Crud, DENTAL_Cru, ACCESS2_Cr, CANCER_Cru, PAPTEST_Cr and COLON_SCRE) being in common between both sets and characterized by consistent effects (OR). In general, individuals within the third tertile of each variables’ distribution were characterized by the greatest impact on clusters distribution. PHLTH_Crud and TEETHLOST were not among the 10 most associated variables identified by the analysis of continuous distributions, but were characterized by consistent OR: subjects with higher values of both variables were significantly more likely to belong to C2, C3 and C4, considering C1 as baseline (OR > 2.5, p-value < 0.001).

A multivariate multinomial logistic regression with a backward stepwise features selection procedure was applied to identify the most informative set of variables, jointly modulating the probability to belong to the clusters. In this case, 20 variables have been selected (Table 4). Of those, five variables have been found to be significant (p < 0.01) in all sub-regressions performed by the multinomial model: colon screening (fecal occult blood test, sigmoidoscopy, or colonoscopy among adults aged 50–75 years), chronic obstructive pulmonary disease among adults aged ≥ 18 years, high cholesterol among adults aged ≥ 18 years who have been screened in the past 5 years, chronic kidney disease among adults aged ≥ 18 years and finally, stroke among adults aged ≥ 18 years. Compared to subjects in C1, individuals within C2, C3 and C4 were characterized by significantly higher values of colon screening, higher cholesterol and stroke levels (OR > 2, p-value < 0.001), while lower values of Chronic obstructive pulmonary disease and chronic kidney disease (OR < 0.25, p-value < 0.01).

When performing multivariate analyses of discretized variables, a total number of 20 features was selected (Table 5): this set of features was partially overlapping with those identified as informative using continuous distributions. Of these variables, CHD (2nd and 3rd tertiles, corresponding to the intervals 5.100–5.883 and ≥ 5.884) and obesity (2nd tertile, corresponding to the interval 23.171–28.853) reached statistical significance for at least one distribution interval for all the three reference clusters (p-value < 0.01). In particular, patients with CHD values within the 2nd and 3rd tertiles were more likely to belong to C2, C3 and C4 compared to C1 (OR > 2, p-value < 0.05). On the opposite, subjects with obesity levels within the 3rd tertile were less likely to be included into C2, C3 and C4 considering C1 as baseline (OR < 0.50, p-value < 0.001).

In general, cluster C1, which is the one that groups green areas, has consistently better prevention and health indicators, but worse sleeping indexes and leisure time. Overall, there is a gradient with all indexes moving from cluster C1, to C2, to C3 and finally to C4, which are the residential areas with large buildings.

### 3.2. Clusters Validation with Inter-Rater Agreement and Cohen’s Kappa Coefficient

In order to further test the reliability of the found clusters in terms of human interpretation, we performed an additional test based on the human-machine agreement. This concept is widely used in artificial intelligence as a measure of reliability of an automated classification, following the idea that, in a certain application, the automated process should be able to perform at least as well as a human being, but in a much smaller time. In our case, we pooled the judgement of six people and asked them to rate a few hundreds of images presented to them one by one. These images were taken from the set of tiles that had already been classified by our algorithm, but the result of the classification was unknown to the human raters. Each rater was asked to associate with each image a number ranging from 1 to 5, indicating:Green areas (parks, gardens, nature);Residential areas with small houses;Industrial areas with factories, storage buildings and construction sites;Highly urbanized areas with large buildings.

Of the 2512 images classified by the algorithm, 1158 were also classified by our raters, so our comparison was performed on approximately 46% of the entire dataset. Figure 15 shows the confusion matrix resulting from this test. In the main diagonal of the matrix it is possible to visualize how many clusters were identified in the same way by both human raters (rows) and the algorithm (columns), on the right side of this matrix; two columns reporting the true positive rates (left column) and the false negative rates (right column) are represented, whereas on the bottom, the positive predictive value (upper row) and the false discovery rate (lower row) are reported. Looking at these results, we can see that the human-machine agreement is generally moderately high, except for cluster 3, representing the industrial areas. In this case, human raters identified as industrial zones only 43.9% of the images identified in the same way by the algorithm, but, in spite of this, 91.6% of these images were areas identified as industrial by the algorithm. This result is not unexpected, since industrial areas can be easily mistaken for highly urbanized residential areas, as we can see that most of the human raters classified those images as belonging to this category. Another possible explanation is that the cluster 3 found by the algorithm could not represent specifically industrial areas, but a sort of buffer zone between residential areas and highly urbanized ones, where health indicators tend to worsen compared to the first ones, but these still do not enter in the cluster 4 range of values.

Finally, we analyzed the results using Cohen’s Kappa coefficient [32], a metric that measures agreement between two raters, also taking into consideration the possibility of the agreement to be occurring by chance. In our case, we obtained Kappa = 0.58, that can be interpreted as moderate agreement, although very close to the substantial agreement threshold, conventionally considered to be 0.6. In particular, a random agreement of 0.29 and a maximum possible Kappa of 0.855 were estimated. Hypothesis testing over the confidence interval of Kappa confirmed these results, rejecting the null hypothesis of random agreement by the two raters with a p-value lower than $1\times 10^{-4}.$

### 3.3. Mapping

The blocks and the clusters have been represented in the original map, confirming the qualitative evaluation of the clusters reported above (Figure 16).

We can find green areas (C1), residential areas with larger buildings (C2), industrial areas (C3) and finally, residential areas (C4).

A curiosity that can be observed in Figure 16 is that there are some areas of New York that are not colored, and they therefore do not belong to any cluster. Some areas in the city are not taken into consideration in the census procedures, since they are mostly uninhabited. Specifically, these areas include the industrial docks, JFK and La Guardia airports, cemeteries and national parks.

## 4. Discussion

Even though in most cases the specific mechanisms are unknown, the influence of socioeconomic status and environmental factors on health is widely recognized [33,34,35]. In our modern society, along with technological development, we have to deal with a new set of exposures that can have an impact on our health often difficult to quantify [36,37]. This is especially true in densely populated areas, where population heterogeneity complicates these kinds of studies.

On the other hand, with the quick development of big data based technology, collaborative systems such as the one developed in PULSE allow us to intervene on specific public health problems more easily, quickly and effectively. PULSE is based on a data exchange cycle where single citizens and policy makers are interconnected, and data coming from the citizens can be used by the policy makers to help the citizens themselves, through a proper intersection of data analysis, risk assessment and personalized notifications.

Furthermore, even in a large heterogeneous environment like a big city it could be hypothesized that areas with similar urban structure share also a similar health status, since it is known that the socioeconomic status of a neighborhood is somehow connected to health and mirrored by the urban structure at some point. If that is the case, this information could be used to design intervention strategies even more quickly, since interventions on a specific neighborhood are expected to have similar effects in other neighborhoods that have the same characteristics.

In this study, we provided proof that correlations between urban structure and health outcomes can be spotted even using something as simple as a satellite image, and we provided an analysis pipeline to find such correlations. We found that in NYC, indubitably one of the most heterogeneous cities in the world, the link between urban landscape and health indicators is particularly strong, and the local presence of green areas such as parks, gardens and nature has an impact on population’s health status.

Looking at the detailed results, it is possible to draw the conclusion that areas with the highest percentage of green are also those in which the population tends to have less health complications because prevention is performed more correctly, and people have healthier habits. It should be noted that this apparently simple link may hide a large number of social implications that are the result of several peculiarities of the US social system, where generally speaking health, education and prevention are highly correlated to income and social class. Even within PULSE, another study demonstrated that poverty rate is a good predictor for asthma hospitalizations [35].

Of course, these results do not necessarily show that green areas do not need any kind of interventions, since by looking at each single factor, some specific criticalities could be spotted, even in the areas where the general health status is good. For instance, concerning the sleep indicator, which reports the percentage of people that declare that they sleep at least 7 h per night, it is possible to see that in the green areas people tend to be more sleep deprived. This could be the result of longer commute times to go to work or even of a more stressful life caused by mentally demanding jobs. It then seems clear that the interventions should be focused on specific variables for specific areas.

Our work has a number of implications.

First of all, it shows that deep neural networks designed to encode image data can be successfully reused within transfer learning approaches. Their application to represent urban landscape seems very effective.

Second, in the context of the PULSE project, the capability of finding clusters of similar urban landscapes may allow one to profile city areas, in which health care decision makers may plan similar interventions.

Finally, the combination of urban landscape and healthcare indicators is not only useful to hypothesize the intertwining of these two dimensions, but also to further profile urban areas, by finding similar areas with similar behaviors of their inhabitants, thus also allowing lifestyle interventions and more “precise” and “personalized” health care policies.

Of course, the analysis has some limitations. First of all, the “quantification” of the health care indexes in the city blocks have been performed by a weighted averaging of the indexes of the census tracts included in the blocks. The weights are computed taking into account only the spatial overlap and not the actual number of inhabitants of the blocks. Although this issue has already partially dealt with, since census tracts are conventionally designed in order to have similar population densities, and tend to be smaller in highly populated areas, some more precise ways to weigh the results on the population of each areas can be tested. Secondly, the results obtained are probably “proxies” of the wealth of the people living in the different areas. For this reason, results may be representative of specific cities and not generalizable to other ones.

## Figures and Tables

**Figure 1 sensors-20-02105-f001:**
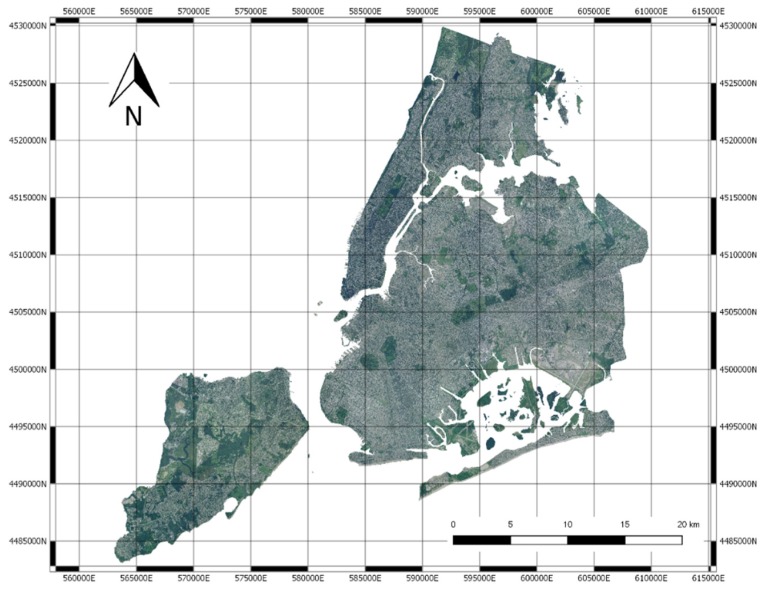
The National Agriculture Imagery Program (NAIP) image of New York City (NYC).

**Figure 2 sensors-20-02105-f002:**
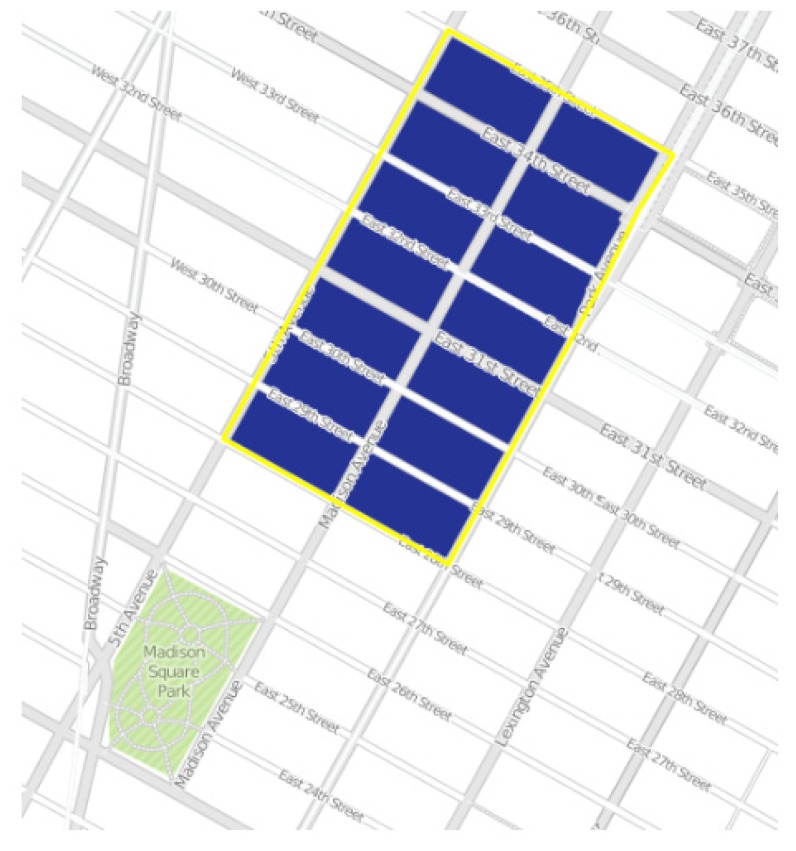
A census tract image of Manhattan.

**Figure 3 sensors-20-02105-f003:**
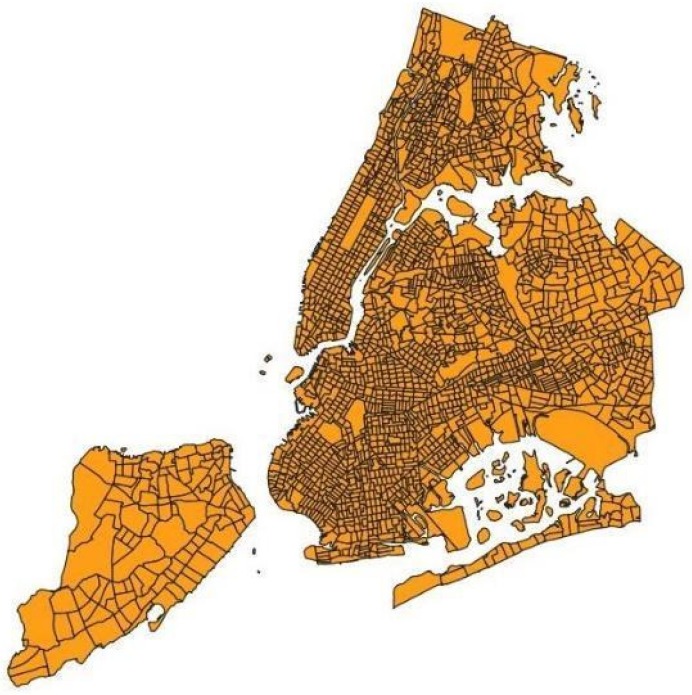
The census tracts of NYC.

**Figure 4 sensors-20-02105-f004:**
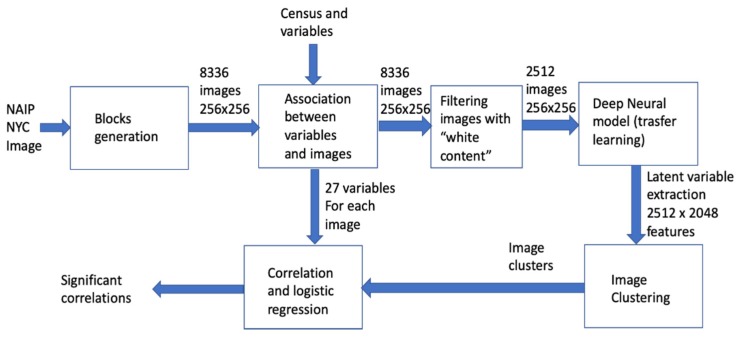
The data analysis pipeline.

**Figure 5 sensors-20-02105-f005:**
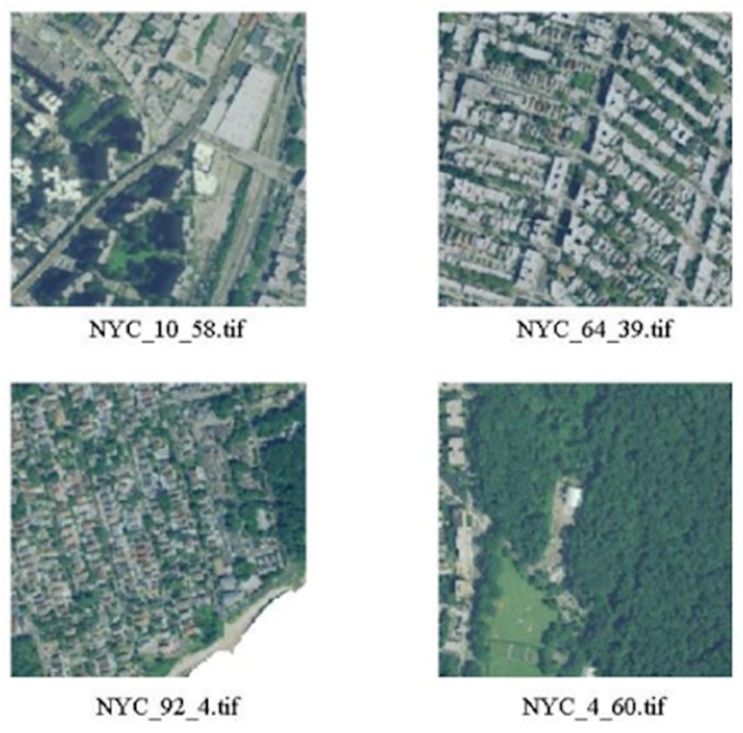
In Figure 5, some of the images obtained from the NAIP images of NY are shown.

**Figure 6 sensors-20-02105-f006:**
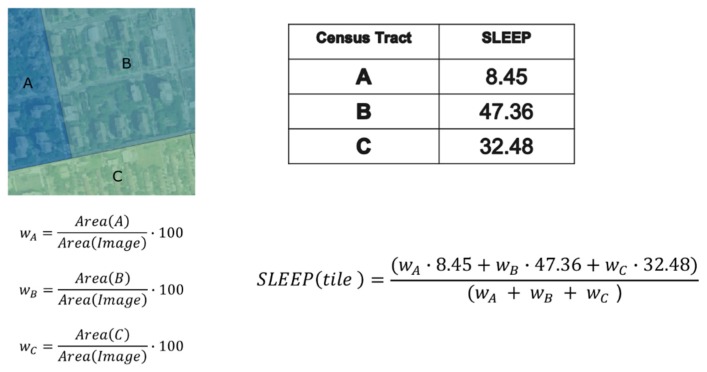
The quantification of the healthcare index value (SLEEP = percentage of people that declares to be sleeping at least 7 h per night on average) of a block.

**Figure 7 sensors-20-02105-f007:**
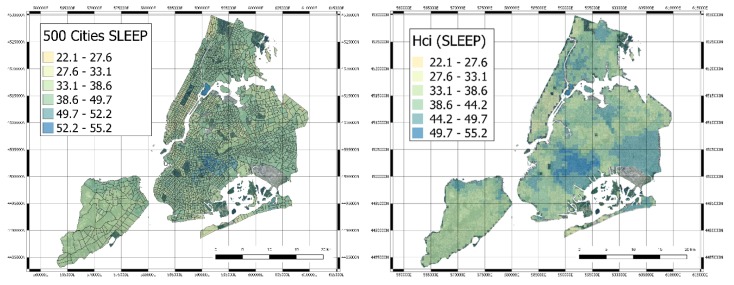
Original Census Tracts with the 500 Cities SLEEP index (left hand side) and derived quantification of the healthcare index values for SLEEP variable (right hand side).

**Figure 8 sensors-20-02105-f008:**
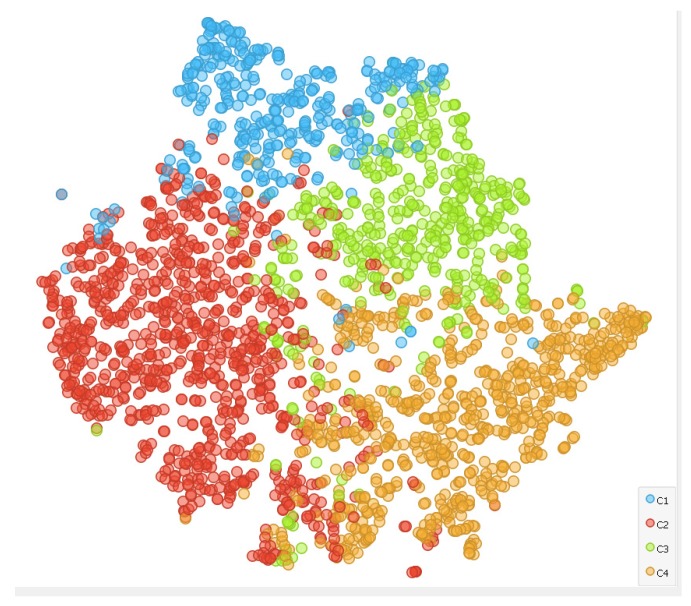
The tSNE representation of the data with colors identifying the four clusters.

**Figure 9 sensors-20-02105-f009:**
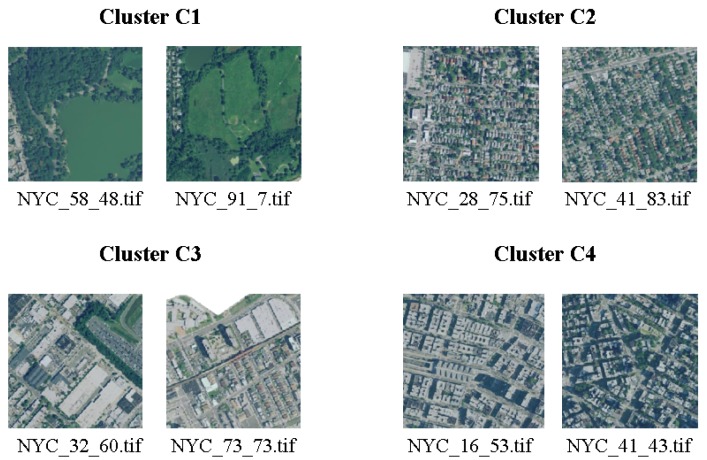
Two images for each cluster.

**Figure 10 sensors-20-02105-f010:**
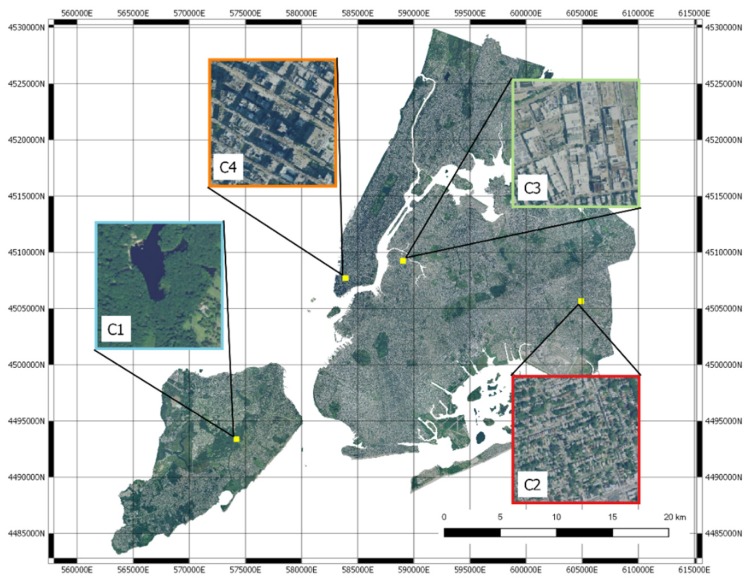
Four images representing the clusters mapped to the NYC map.

**Figure 11 sensors-20-02105-f011:**
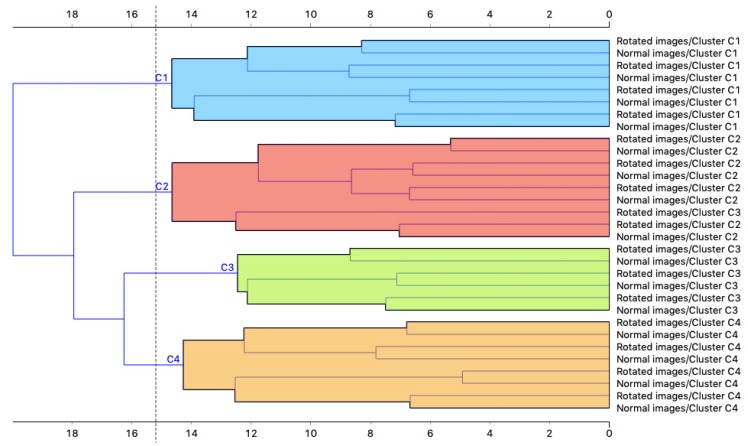
Dendrogram resulting from the hierarchical clustering of original and rotated images.

**Figure 12 sensors-20-02105-f012:**
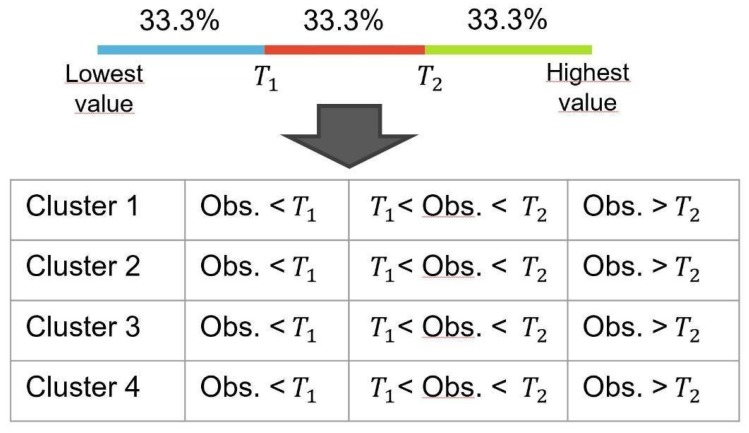
Discretization of the health variables and creation of the contingency tables for the Chi-squared test.

**Figure 13 sensors-20-02105-f013:**
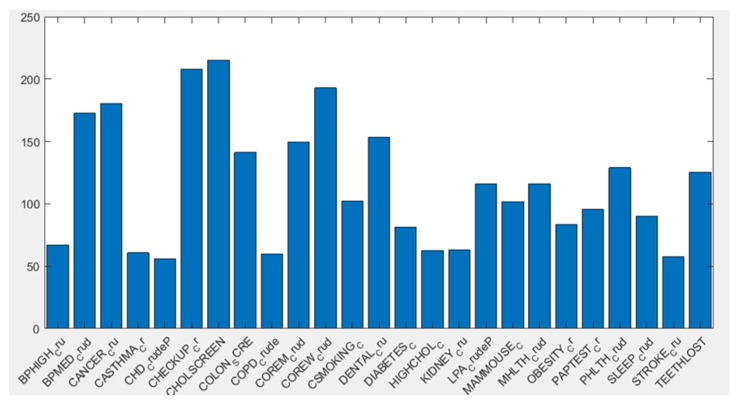
Negative logarithm of the p-value resulting from the Chi-squared test for each one of the variables.

**Figure 14 sensors-20-02105-f014:**
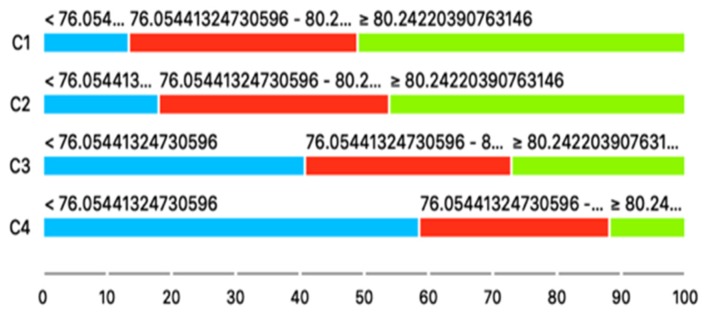
The different distributions of Cholesterol screening among adults aged ≥ 18 years in the different clusters. Inhabitants of cluster C1 have much higher propensity towards screening than those who live in Cluster C4.

**Figure 15 sensors-20-02105-f015:**
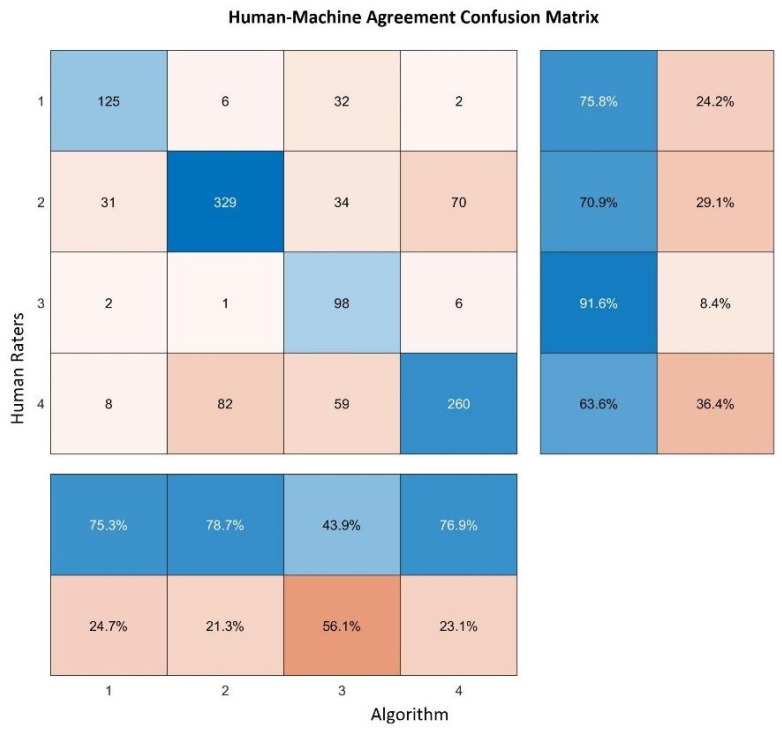
Confusion matrix of the results of our human-machine comparison.

**Figure 16 sensors-20-02105-f016:**
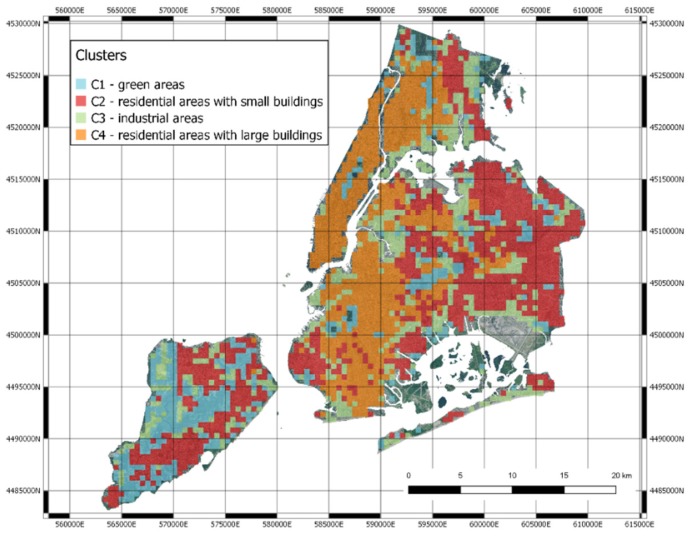
The clusters remapped in NYC.

**Table 1 sensors-20-02105-t001:** 500 cities measures grouped by category. The 24 measures include 13 health outcomes, 9 prevention practices and 5 unhealthy behaviors.

Category	Measure
Health outcomes	Current asthma among adults aged >= 18 years High blood pressure among adults aged >= 18 years Cancer among adults aged ≥ 18 years High cholesterol among adults aged >= 18 years who have been screened in the past 5 years Chronic kidney disease among adults aged ≥ 18 years Chronic obstructive pulmonary disease among adults aged >= 18 years Coronary heart disease among adults aged ≥ 18 years Diagnosed diabetes among adults aged >= 18 years Mental health not good for >= 14 days among adults aged >= 18 years Physical health not good for >= 14 days among adults aged >= 18 years All teeth lost among adults aged >= 65 years Stroke among adults aged >= 18 years
Prevention	Visits to doctor for routine checkup within the past year among adults aged ≥ 18 years Visits to dentist or dental clinic among adults aged ≥ 18 years Taking medicine for high blood pressure control among adults aged ≥ 18 years with high blood pressure Cholesterol screening among adults aged ≥ 18 years Mammography use among women aged 50−74 years Papanicolaou smear use among adult women aged 21−65 years Fecal occult blood test, sigmoidoscopy, or colonoscopy among adults aged 50–75 years Older adults aged ≥ 65 years who are up to date on a core set of clinical preventive services by age and sex
Unhealthy behaviors	Current smoking among adults aged >= 18 years No leisure-time physical activity among adults aged >= 18 years Obesity among adults aged >= 18 years Sleeping less than 7 h among adults aged >= 18 years

**Table 2 sensors-20-02105-t002:** Results from univariate multinomial logistic regression using continuous variables. Variable = variable included in the model; OR = odds ratio expressing the risk to belong to each reference cluster compared to the risk to belong to the baseline cluster C1 by 1 unit increase of each variable; SE = standard error of the regression coefficient (beta); p = p-value expressing the probability to observe “by chance” a difference in terms of variables’ distribution between the reference classes compared to the baseline value greater than the effect estimated from data. The null hypothesis is that variables’ distribution is the same in each reference cluster compared to the baseline.

Variable	C2 vs. C1		C3 vs. C1		C4 vs. C1	
	OR (SE)	p	OR (SE)	p	OR (SE)	p
ACCESS2_Cr	1.07 (0.01)	<0.001	1.15 (0.01)	<0.001	1.17 (0.01)	<0.001
BINGE_Crud	0.81 (0.02)	<0.001	0.91 (0.02)	<0.001	1.01 (0.02)	0.577
BPHIGH_Cru	1.06 (0.01)	<0.001	1.04 (0.01)	0.004	0.98 (0.01)	0.092
BPMED_Crud	1.09 (0.02)	<0.001	0.91 (0.02)	<0.001	0.87 (0.02)	<0.001
CANCER_Cru	0.86 (0.04)	<0.001	0.6 (0.05)	<0.001	0.47 (0.05)	<0.001
CASTHMA_Cr	1.35 (0.05)	<0.001	1.61 (0.05)	<0.001	1.77 (0.05)	<0.001
CHD_CrudeP	1.04 (0.04)	0.374	0.98 (0.05)	0.660	0.88 (0.05)	0.005
CHECKUP_Cr	1.08 (0.02)	<0.001	0.94 (0.02)	0.001	0.8 (0.02)	<0.001
CHOLSCREEN	0.95 (0.02)	0.001	0.82 (0.02)	<0.001	0.74 (0.02)	<0.001
COLON_SCRE	0.97 (0.01)	0.002	0.9 (0.01)	<0.001	0.88 (0.01)	<0.001
COPD_Crude	1.17 (0.04)	<0.001	1.25 (0.05)	<0.001	1.18 (0.04)	<0.001
COREM_Crud	0.88 (0.02)	<0.001	0.81 (0.02)	<0.001	0.77 (0.02)	<0.001
COREW_Crud	0.88 (0.01)	<0.001	0.82 (0.01)	<0.001	0.77 (0.01)	<0.001
CSMOKING_C	1.06 (0.02)	0.001	1.18 (0.02)	<0.001	1.18 (0.02)	<0.001
DENTAL_Cru	0.94 (0.01)	<0.001	0.9 (0.01)	<0.001	0.9 (0.01)	<0.001
DIABETES_C	1.18 (0.02)	<0.001	1.25 (0.03)	<0.001	1.24 (0.02)	<0.001
HIGHCHOL_C	1.02 (0.02)	0.306	0.92 (0.02)	<0.001	0.86 (0.02)	<0.001
KIDNEY_Cru	2.18 (0.14)	<0.001	2.82 (0.16)	<0.001	2.96 (0.15)	<0.001
X.LPA_CrudeP	1.09 (0.01)	<0.001	1.14 (0.01)	<0.001	1.14 (0.01)	<0.001
MAMMOUSE_C	0.98 (0.02)	0.185	0.9 (0.02)	<0.001	0.83 (0.02)	<0.001
MHLTH_Crud	1.19 (0.03)	<0.001	1.41 (0.03)	<0.001	1.47 (0.03)	<0.001
OBESITY_Cr	1 (0.01)	0.636	1.06 (0.01)	<0.001	1.02 (0.01)	0.084
PAPTEST_Cr	0.85 (0.02)	<0.001	0.84 (0.02)	<0.001	0.81 (0.02)	<0.001
PHLTH_Crud	1.15 (0.02)	<0.001	1.28 (0.02)	<0.001	1.31 (0.02)	<0.001
SLEEP_Crud	1.15 (0.01)	<0.001	1.2 (0.02)	<0.001	1.19 (0.02)	<0.001
STROKE_Cru	1.51 (0.07)	<0.001	1.64 (0.08)	<0.001	1.61 (0.08)	<0.001
TEETHLOST	1.09 (0.01)	<0.001	1.15 (0.01)	<0.001	1.18 (0.01)	<0.001

**Table 3 sensors-20-02105-t003:** Results from univariate multinomial logistic regression using discretized variables. Variable = variable included in the model; Interval = variables’ interval (the first tertile was used as intercept of the model); OR = odds ratio; SE = standard error of the regression coefficient (beta); p = p-value.

Variable	Interval	C2 vs. C1		C3 vs. C1		C4 vs. C1	
		OR (SE)	p	OR (SE)	p	OR (SE)	p
ACCESS2_Cr	12–18.95	2.6 (0.14)	<0.001	3.41 (0.17)	<0.001	2.93 (0.16)	<0.001
ACCESS2_Cr	>=18.95	3.05 (0.18)	<0.001	10.48 (0.2)	<0.001	11.33 (0.19)	<0.001
BINGE_Crud	13.98–16.73	0.72 (0.16)	0.045	1.05 (0.18)	0.776	1.19 (0.17)	0.306
BINGE_Crud	>=16.73	0.25 (0.15)	<0.001	0.48 (0.17)	<0.001	0.73 (0.16)	0.044
BPHIGH_Cru	28.4–32.9	3.65 (0.15)	<0.001	1.37 (0.17)	0.064	0.92 (0.16)	0.590
BPHIGH_Cru	>=32.9	2.2 (0.15)	<0.001	1.43 (0.16)	0.027	1.1 (0.15)	0.530
BPMED_Crud	73.15–76.47	2.52 (0.17)	<0.001	0.69 (0.17)	0.028	0.59 (0.15)	0.001
BPMED_Crud	>=76.47	2.84 (0.16)	<0.001	0.59 (0.17)	0.001	0.21 (0.17)	<0.001
CANCER_Cru	4.7–5.95	1.71 (0.18)	0.002	0.51 (0.18)	<0.001	0.36 (0.17)	<0.001
CANCER_Cru	>=5.95	1.05 (0.16)	0.769	0.24 (0.18)	<0.001	0.11 (0.17)	<0.001
CASTHMA_Cr	9.1–10.5	1.28 (0.14)	0.073	1.53 (0.16)	0.009	1.21 (0.15)	0.204
CASTHMA_Cr	>=10.5	2.48 (0.17)	<0.001	4.69 (0.19)	<0.001	5.23 (0.18)	<0.001
CHD_CrudeP	5.1–5.88	3.53 (0.15)	<0.001	1.28 (0.17)	0.146	1.2 (0.16)	0.257
CHD_CrudeP	>=5.88	2.17 (0.15)	<0.001	1.27 (0.16)	0.142	1.57 (0.15)	0.003
CHECKUP_Cr	73.56–76.29	1.07 (0.17)	0.686	0.32 (0.18)	<0.001	0.16 (0.16)	<0.001
CHECKUP_Cr	>=76.29	2.05 (0.18)	<0.001	0.53 (0.18)	<0.001	0.19 (0.18)	<0.001
CHOLSCREEN	76.05–80.24	0.75 (0.19)	0.125	0.3 (0.2)	<0.001	0.19 (0.18)	<0.001
CHOLSCREEN	>=80.24	0.67 (0.18)	0.028	0.17 (0.19)	<0.001	0.05 (0.2)	<0.001
COLON_SCRE	60.09–66.4	0.45 (0.19)	<0.001	0.23 (0.2)	<0.001	0.11 (0.19)	<0.001
COLON_SCRE	>=66.4	0.56 (0.19)	0.002	0.18 (0.2)	<0.001	0.12 (0.19)	<0.001
COPD_Crude	5.47–6.41	1.82 (0.14)	<0.001	1.44 (0.16)	0.025	0.67 (0.16)	0.010
COPD_Crude	>=6.41	1.74 (0.16)	<0.001	2.07 (0.17)	<0.001	2.28 (0.16)	<0.001
COREM_Crud	25.43–31.74	0.61 (0.21)	0.022	0.44 (0.22)	<0.001	0.25 (0.21)	<0.001
COREM_Crud	>=31.74	0.27 (0.2)	<0.001	0.1 (0.22)	<0.001	0.06 (0.2)	<0.001
COREW_Crud	24.12–29.59	0.88 (0.21)	0.552	0.4 (0.22)	<0.001	0.25 (0.21)	<0.001
COREW_Crud	>=29.59	0.27 (0.19)	<0.001	0.11 (0.2)	<0.001	0.06 (0.2)	<0.001
CSMOKING_C	15.1–17.96	1.07 (0.13)	0.596	1.84 (0.17)	<0.001	0.76 (0.15)	0.078
CSMOKING_C	>=17.96	1.83 (0.17)	0.001	5.3 (0.19)	<0.001	4.97 (0.18)	<0.001
DENTAL_Cru	58.83–69.13	0.81 (0.2)	0.288	0.39 (0.21)	<0.001	0.27 (0.2)	<0.001
DENTAL_Cru	>=69.13	0.26 (0.18)	<0.001	0.1 (0.2)	<0.001	0.09 (0.19)	<0.001
DIABETES_C	9.11–11.64	2.3 (0.14)	<0.001	2.52 (0.16)	<0.001	1.36 (0.15)	0.050
DIABETES_C	>=11.64	3.09 (0.17)	<0.001	4.65 (0.19)	<0.001	5.2 (0.17)	<0.001
HIGHCHOL_C	36.22–39.25	2.16 (0.16)	<0.001	0.9 (0.17)	0.556	0.71 (0.16)	0.032
HIGHCHOL_C	>=39.25	1.59 (0.15)	0.002	0.62 (0.17)	0.004	0.43 (0.15)	<0.001
KIDNEY_Cru	1.95–2.35	2.97 (0.15)	<0.001	2.28 (0.17)	<0.001	1.72 (0.16)	0.001
KIDNEY_Cru	>=2.35	1.83 (0.15)	<0.001	2.43 (0.17)	<0.001	3.06 (0.15)	<0.001
X.LPA_CrudeP	23.99–30.5	2.93 (0.14)	<0.001	2.5 (0.17)	<0.001	2.14 (0.16)	<0.001
X.LPA_CrudeP	>=30.5	3.41 (0.18)	<0.001	7.28 (0.19)	<0.001	7.36 (0.18)	<0.001
MAMMOUSE_C	74.24–76.74	0.73 (0.17)	0.055	0.47 (0.18)	<0.001	0.18 (0.17)	<0.001
MAMMOUSE_C	>=76.74	0.99 (0.17)	0.929	0.49 (0.18)	<0.001	0.26 (0.17)	<0.001
MHLTH_Crud	10.8–13.01	1.49 (0.13)	0.003	2.81 (0.17)	<0.001	1.45 (0.15)	0.017
MHLTH_Crud	>=13.01	2.23 (0.18)	<0.001	7.3 (0.2)	<0.001	7.23 (0.18)	<0.001
OBESITY_Cr	23.17–28.85	0.53 (0.14)	<0.001	0.69 (0.17)	0.034	0.24 (0.16)	<0.001
OBESITY_Cr	>=28.85	1.26 (0.18)	0.184	2.83 (0.19)	<0.001	1.44 (0.18)	0.037
PAPTEST_Cr	79.76–84.2	0.44 (0.2)	<0.001	0.35 (0.21)	<0.001	0.2 (0.2)	<0.001
PAPTEST_Cr	>=84.2	0.2 (0.19)	<0.001	0.18 (0.2)	<0.001	0.1 (0.19)	<0.001
PHLTH_Crud	10.68–13.24	2 (0.14)	<0.001	2.07 (0.17)	<0.001	1.39 (0.16)	0.035
PHLTH_Crud	>=13.24	2.65 (0.18)	<0.001	6.8 (0.19)	<0.001	7.83 (0.18)	<0.001
SLEEP_Crud	37.6–42.92	1.56 (0.14)	0.001	4.11 (0.17)	<0.001	2.24 (0.15)	<0.001
SLEEP_Crud	>=42.92	3.96 (0.18)	<0.001	9.75 (0.21)	<0.001	7.59 (0.19)	<0.001
STROKE_Cru	2.49–3.2	3.13 (0.15)	<0.001	1.9 (0.17)	<0.001	1.66 (0.16)	0.001
STROKE_Cru	>=3.2	2.1 (0.15)	<0.001	2.22 (0.17)	<0.001	2.87 (0.15)	<0.001
TEETHLOST	12.59–17.3	1.49 (0.13)	0.003	2.09 (0.16)	<0.001	1 (0.16)	0.982
TEETHLOST	>=17.3	2.81 (0.19)	<0.001	6.38 (0.2)	<0.001	8.2 (0.19)	<0.001

**Table 4 sensors-20-02105-t004:** Results from multivariate multinomial logistic regression using continuous variables. Variable = variable included in the model; OR = odds ratio; SE = standard error of the regression coefficient (beta); p = p-value from multivariate multinomial logistic regression.

Variable	C2 vs. C1		C3 vs. C1		C4 vs. C1	
	OR (SE)	p	OR (SE)	p	OR (SE)	p
BPMED_Crud	1.08 (0.12)	0.524	0.65 (0.11)	<0.001	0.81 (0.11)	0.046
CANCER_Cru	0.29 (0.4)	0.002	0.83 (0.41)	0.658	0.84 (0.4)	0.664
CASTHMA_Cr	0.3 (0.48)	0.012	0.58 (0.46)	0.232	7.3 (0.46)	<0.001
CHD_CrudeP	15.53 (0.61)	<0.001	1.18 (0.63)	0.788	1.21 (0.63)	0.764
CHECKUP_Cr	0.73 (0.19)	0.095	0.72 (0.19)	0.083	0.38 (0.17)	<0.001
COLON_SCRE	1.89 (0.1)	<0.001	1.89 (0.1)	<0.001	2.1 (0.12)	<0.001
COPD_Crude	0.12 (0.53)	<0.001	0.21 (0.5)	0.002	0.02 (0.53)	<0.001
COREM_Crud	0.93 (0.12)	0.582	1.34 (0.13)	0.026	1.08 (0.14)	0.574
COREW_Crud	1.1 (0.11)	0.414	0.74 (0.12)	0.012	0.54 (0.13)	<0.001
CSMOKING_C	0.42 (0.24)	<0.001	0.91 (0.22)	0.662	0.7 (0.23)	0.124
HIGHCHOL_C	2.8 (0.17)	<0.001	2.2 (0.19)	<0.001	3.24 (0.2)	<0.001
KIDNEY_Cru	0 (0.76)	<0.001	0.09 (0.75)	0.001	0 (0.77)	<0.001
X.LPA_CrudeP	1.2 (0.12)	0.116	1.37 (0.11)	0.006	0.99 (0.11)	0.925
MAMMOUSE_C	0.91 (0.11)	0.396	0.65 (0.11)	<0.001	0.57 (0.11)	<0.001
MHLTH_Crud	95.21 (0.52)	<0.001	5.03 (0.48)	0.001	1.94 (0.51)	0.195
OBESITY_Cr	0.87 (0.05)	0.004	1.01 (0.05)	0.854	0.81 (0.05)	<0.001
PHLTH_Crud	0.08 (0.52)	<0.001	0.29 (0.47)	0.008	0.62 (0.48)	0.324
SLEEP_Crud	1.59 (0.16)	0.004	1.43 (0.17)	0.031	1.32 (0.16)	0.079
STROKE_Cru	6.86 (0.59)	0.001	42.51 (0.57)	<0.001	384.21 (0.61)	<0.001
TEETHLOST	1.93 (0.15)	<0.001	1.13 (0.15)	0.428	1.56 (0.15)	0.004

**Table 5 sensors-20-02105-t005:** Results from multivariate multinomial logistic regression using discretized variables. Variable = variable included in the model and corresponding tertile; OR = odds ratio; SE = standard error of the regression coefficient (beta); p-value = p-value from multivariate multinomial logistic regression.

Variable	C2 vs. C1		C3 vs. C1		C4 vs. C1	
	OR (SE)	p	OR (SE)	p	OR (SE)	p
BPHIGH_Cru2	1.02 (0.29)	0.947	1.31 (0.33)	0.422	1.17 (0.34)	0.635
BPHIGH_Cru3	0.24 (0.45)	0.001	0.43 (0.51)	0.100	0.45 (0.53)	0.130
BPMED_Crud2	0.97 (0.34)	0.919	0.46 (0.36)	0.032	0.55 (0.35)	0.086
BPMED_Crud3	1.9 (0.46)	0.165	0.59 (0.5)	0.293	0.48 (0.5)	0.144
CANCER_Cru2	1.43 (0.34)	0.294	0.94 (0.35)	0.855	0.86 (0.34)	0.670
CANCER_Cru3	0.65 (0.48)	0.370	0.74 (0.52)	0.569	0.61 (0.5)	0.318
CASTHMA_Cr2	1.2 (0.24)	0.432	0.82 (0.28)	0.483	1.15 (0.3)	0.628
CASTHMA_Cr3	1.4 (0.49)	0.490	1.45 (0.52)	0.480	6.17 (0.55)	0.001
CHD_CrudeP2	2.69 (0.25)	<0.001	2.08 (0.29)	0.011	3.54 (0.3)	<0.001
CHD_CrudeP3	3.13 (0.38)	0.003	5.07 (0.43)	<0.001	15.13 (0.44)	<0.001
CHECKUP_Cr2	1.32 (0.32)	0.392	0.79 (0.34)	0.471	0.42 (0.33)	0.009
CHECKUP_Cr3	2.61 (0.43)	0.025	1.15 (0.47)	0.767	0.24 (0.47)	0.002
CHOLSCREEN2	0.71 (0.34)	0.320	0.54 (0.36)	0.088	0.77 (0.35)	0.450
CHOLSCREEN3	0.44 (0.45)	0.066	0.31 (0.49)	0.016	0.24 (0.48)	0.003
COLON_SCRE2	1.59 (0.36)	0.200	1.75 (0.38)	0.138	0.77 (0.38)	0.489
COLON_SCRE3	7.47 (0.44)	<0.001	5.42 (0.49)	0.001	2.04 (0.5)	0.153
COPD_Crude2	2.36 (0.25)	0.001	1.99 (0.29)	0.016	0.98 (0.3)	0.954
COPD_Crude3	2.85 (0.36)	0.004	2.16 (0.41)	0.061	1.16 (0.42)	0.730
COREM_Crud2	0.84 (0.38)	0.640	0.87 (0.39)	0.728	0.32 (0.41)	0.005
COREM_Crud3	0.68 (0.53)	0.463	0.57 (0.58)	0.330	0.3 (0.59)	0.040
COREW_Crud2	0.66 (0.4)	0.308	0.58 (0.42)	0.193	0.23 (0.43)	0.001
COREW_Crud3	0.66 (0.55)	0.453	0.81 (0.61)	0.734	0.15 (0.6)	0.002
DENTAL_Cru2	1.15 (0.42)	0.733	0.79 (0.44)	0.595	1.73 (0.45)	0.219
DENTAL_Cru3	0.56 (0.54)	0.292	0.3 (0.59)	0.044	0.6 (0.62)	0.407
HIGHCHOL_C2	0.96 (0.28)	0.876	0.77 (0.29)	0.371	0.41 (0.29)	0.003
HIGHCHOL_C3	0.52 (0.41)	0.112	0.59 (0.44)	0.221	0.23 (0.44)	0.001
KIDNEY_Cru2	0.95 (0.24)	0.843	0.87 (0.28)	0.635	1.14 (0.29)	0.657
KIDNEY_Cru3	0.37 (0.42)	0.017	0.49 (0.47)	0.135	1.77 (0.48)	0.233
X.LPA_CrudeP2	1.12 (0.33)	0.726	0.81 (0.38)	0.593	1.26 (0.4)	0.569
X.LPA_CrudeP3	1.33 (0.5)	0.571	1.11 (0.55)	0.843	0.56 (0.57)	0.305
MHLTH_Crud2	1.16 (0.25)	0.555	1.95 (0.31)	0.034	1.24 (0.34)	0.529
MHLTH_Crud3	0.55 (0.47)	0.212	0.76 (0.53)	0.613	0.6 (0.57)	0.372
OBESITY_Cr2	0.45 (0.25)	0.001	0.35 (0.29)	<0.001	0.07 (0.3)	<0.001
OBESITY_Cr3	0.37 (0.45)	0.024	0.39 (0.48)	0.053	0.04 (0.5)	<0.001
PAPTEST_Cr2	0.59 (0.31)	0.083	0.86 (0.33)	0.655	0.76 (0.33)	0.408
PAPTEST_Cr3	0.24 (0.4)	<0.001	0.65 (0.44)	0.327	0.7 (0.45)	0.428
SLEEP_Crud2	1 (0.3)	0.992	2.18 (0.35)	0.026	0.38 (0.37)	0.008
SLEEP_Crud3	4.26 (0.5)	0.004	4.45 (0.54)	0.006	0.54 (0.55)	0.260

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
