# Peer review of "Deep Learning to Unveil Correlations between Urban Landscape and Population Health†"

_sensors, 2020, doi:10.3390/s20072105_

Round 1
Reviewer 1 Report
Major Remarks
This is an interesting paper, well written and with a nice flow. Nevertheless, there are some important points in the methodology that need to be clarified. From my understanding, the ANN is used to create a kind of land use/cover map. To do so images are clustered based on their percentage of the green color. Authors should make clear why the select this approach instead of using readily available land use/ land cover products or just Normalized Difference Vegetation Index (NDVI) images that quantify vegetation. I think that using an existing LULC map or just an NDVI image would be more reliable. When it comes to satellite images there are more accurate methods than just a ‘visual inspection’ as they mention at line 304. Authors also utilize a method (inter-rater) agreement (line 390) which although used in AI applications, it is rare in Land Use classification mostly due to the fact that it is not as accurate as other methods.
It would be for example more interesting to compare ANN results with an existing LULC product to evaluate the deep learning approach. In fact, if possible, the authors should do that and present the results. If their land use map is more accurate then there would be value in their approach. Otherwise, they should explain (with proper citations) why ANN is better than a casual LULC product.
Furthermore, the results of the multinominal logistic regression are not explained/summarized as they should be. For example, in the current version, only four lines (386-389) are used to explain the results and a couple of lines in the Discussion section. It should be mentioned which variables are correlated (positively or negatively) with which clusters and also further discuss the outcomes from the policy perspective.
Minor comments
Line 101. As authors only refer to one study that applies deep learning to link environment and health, they should add here some more works.
Line 117. It would be useful here to also include a recent review on deep learning applications in various urban analysis problems which also offers technical details for deep learning algorithms.
Grekousis G. 2019. Artificial Neural Networks and Deep Learning in Urban Geography: A systematic review and meta-analysis. Computers, Environment and Urban Systems, 74,244-256
Author Response
We thank the reviewer for the appreciation of our work and the useful comments. We tried to address every part carefully and in the best way as allowed by the time constraints. A detailed response of all the comments can be found below.
Reviewer’s original comment: “From my understanding, the ANN is used to create a kind of land use/cover map. To do so images are clustered based on their percentage of the green color. Authors should make clear why the select this approach instead of using readily available land use/ land cover products or just Normalized Difference Vegetation Index (NDVI) images that quantify vegetation. I think that using an existing LULC map or just an NDVI image would be more reliable.”
Answer: We do not cluster the images based on their percentage of green color. In fact, the green color is not used as a feature to perform clustering, but only as an “external” measure to evaluate the different deep learning architectures in terms of their capability of clustering images. For this reason, we did not consider the NDVI index as the proper clustering indicator, since the level of vegetation and its healthiness are not our focus. Furthermore, NDVI has to be calculated using the infrared band, that we did not have at our disposal. We understand that this point was unclear in the paper, so we clarified this in in lines 293-295.
Reviewer’s original comment: “It would be for example more interesting to compare ANN results with an existing LULC product to evaluate the deep learning approach. In fact, if possible, the authors should do that and present the results. If their land use map is more accurate then there would be value in their approach. Otherwise, they should explain (with proper citations) why ANN is better than a casual LULC product.”
Answer: Land Use and Land Cover analyses are performed with many different methods, most of them based on image analysis machine learning algorithms. Some typical examples are Support Vector Machines and Random Forest. In the time we have at our disposal, it is not easy to perform a complete performance comparison of our neural network with other “traditional” LULC methods, but we added a few lines to the paper (262-271) with some proper citations to explain why we think that the Deep Learning approach works. The main key point stands in the fact that most LULC classification methods are pixel-based, whereas we took a tile-based approach, since our idea was to associate latent variable extracted by pre-trained machine learning models to images representing the urban structure of the city. In image analysis, CNNs are well known for their high accuracy and several studies in literature (some of which we cited in our added text) demonstrate that they generally outperform other methods.
Reviewer’s original comment: “Furthermore, the results of the multinominal logistic regression are not explained/summarized as they should be. For example, in the current version, only four lines (386-389) are used to explain the results and a couple of lines in the Discussion section. It should be mentioned which variables are correlated (positively or negatively) with which clusters and also further discuss the outcomes from the policy perspective”
Answer: We thank the reviewer for highlighting this weak aspect of the manuscript. In order to allow readers for a better interpretation of the results from multinomial logistic regression we included odds ratio and standard error estimates in each table and discussed the results of the most associated variables deriving from each analysis. The whole section (lines 380-441) was expanded and the description improved.
Reviewer’s original comment: “Line 101. As authors only refer to one study that applies deep learning to link environment and health, they should add here some more works.
Line 117. It would be useful here to also include a recent review on deep learning applications in various urban analysis problems which also offers technical details for deep learning algorithms.
Grekousis G. 2019. Artificial Neural Networks and Deep Learning in Urban Geography: A systematic review and meta-analysis. Computers, Environment and Urban Systems, 74,244-256”
Answer: we cited three different works (lines 91-102), one of which is a systematic review reporting many studies on this topic. We thank the reviewer for suggesting another citation, we added it in line 123.
Reviewer 2 Report
Major problem:
Good paper, but I have questions about sensitivity on image rotation.
Rotation is not mentioned as an augmentation method for deep learning, so image segmentation could be sensitive on rotation. Please extend and clarify this part.
The suggested approach is the random selection of areas (ROIs), multiple random rotation of ROIs and the comparison of segmentation results with some criteria.
Author Response
We thank the reviewer for the appreciation of our work and for raising an interesting point. It is true that image rotation can represent a complication for some deep learning algorithms, as some examples in literature show, in particular in presence of adversarial rotation. In our case, given that our CNN does not have the aim to recognize shapes and figures inside the images, we assumed that image rotation would not represent a problem. Anyway, to confirm our theory, we added a section of the paper (lines 331-340 and Figure 11) explaining a little additional experiment we performed, in which we rotated some images and re-clustered them in order to see whether the clustering results were different. This is the most likely situation that may occur when we cluster image tiles. Our results show that in our case these kinds of image rotation do not influence the outcome of the algorithm.
Round 2
Reviewer 1 Report
Most of my comments have been successfully addressed taking into account the time constraints. A minor comment: citation 14 is published in 2019.
Reviewer 2 Report
ok